# ROLE OF TWO LEARNING RATES IN CONVERGENCE OF MODEL-AGNOSTIC META-LEARNING

## ABSTRACT

Model-agnostic meta-learning (MAML) is known as a powerful meta-learning method. However, MAML is notorious for being hard to train because of the existence of two learning rates. Therefore, in this paper, we derived a sufficient condition of inner learning rate $\alpha$ and meta-learning rate $\beta$ for a simplified MAML to locally converge to local minima from any point in the vicinity of the local minima. We find that the upper bound of $\beta$ depends on $\alpha$. Moreover, we show that the threshold of $\beta$ increases as $\alpha$ approaches its own upper bound. This result is verified by experiments on various few-shot tasks and architectures; specifically, we perform sinusoid regression and classification of Omniglot and MiniImagenet datasets with a multilayer perceptron and a convolutional neural network. Based on this outcome, we present a guideline for determining the learning rates: first, search for the largest possible $\alpha$; next, tune $\beta$ based on the chosen value of $\alpha$.

## 1 INTRODUCTION

A pillar of human intelligence is the ability to learn and adapt to unseen tasks quickly and based on only a limited quantity of data. Although machine learning has achieved remarkable results, many recent models require massive quantities of data and are designed for solving particular tasks. Meta-learning, one of the ways of tackling this problem, tries to develop a model that can adapt to new tasks quickly by learning to learn new concepts from few data points (Schmidhuber, 1987; Thrun & Pratt, 1998).

Among meta-learning algorithms, model-agnostic meta-learning (MAML), a gradient-based meta-learning method proposed by Finn et al. (2017), has recently been extensively studied. For example, MAML is used for continual learning (Finn et al., 2019; Jerfel et al., 2019; Spigler, 2019; Al-Shedivat et al., 2018), reinforcement learning (Finn et al., 2017; Al-Shedivat et al., 2018; Gupta et al., 2018; Deleu & Bengio, 2018; Liu & Theodorou, 2019) and probablistic inference (Finn et al., 2018; Yoon et al., 2018; Grant et al., 2018). The reason why MAML is widely used is because MAML is simple but efficient and applicable to a wide range of tasks independent of model architecture and the loss function. However, MAML is notorious for being hard to train (Antoniou et al., 2019). One of the reasons why training MAML is hard is the existence of two learning rates in MAML: the inner learning rate $\alpha$ and meta-learning rate $\beta$. A learning rate is known to be one of the most important parameters, and tuning this parameter may be challenging even if the simple gradient descent (GD) method is used. Nevertheless, we do not yet know the relationship between these two learning rates and have little guidance on how to tune them. Hence, guidelines for choosing these parameters are urgently needed.

In this paper, we investigate the MAML algorithm and propose a guideline for selecting the learning rates. First, in Section 2 we briefly explain by using an approximation how MAML can be regarded as optimization with the negative gradient penalty. Because the gradient norm is related to the shape of the loss surface, a bias towards a larger gradient norm can make training unstable. Next, based on the approximation explained in Section 2, in Section 3, we derive a sufficient condition of $\alpha$ and $\beta$ for a simplified MAML to locally converge to local minima from any point in the neighborhood of the local minima. Furthermore, by removing a constraint, we derive a sufficient condition for local convergence with fewer simplifications as well. We find that the upper bound $\beta_c$ of meta-learning rate depends on inner learning rate $\alpha$. In particular, $\beta_c$ of $\alpha \approx \alpha_c$ is larger than that of $\alpha = 0$,

where $\alpha_c$ is the upper bound of $\alpha$. This is verified by experiments in Section 5. These results imply a guideline for selecting the learning rates: first, search for the largest possible $\alpha$; next, tune $\beta$.

## 2 MAML AS OPTIMIZATION WITH NEGATIVE GRADIENT PENALTY

### 2.1 MAML

The goal of MAML is to find a representation that can rapidly adapt to new tasks with a small quantity of data. In other words, MAML performs optimization for parameters $\boldsymbol{\theta} \in \mathbb{R}^d$ that the optimizer can use to quickly reach the optimal parameter $\boldsymbol{\theta}_\tau^*$ for task $\tau$ with few data points. To this end, MAML takes the following steps to update $\boldsymbol{\theta}$. First, it samples a batch of tasks from task distribution $P(\tau)$ and updates $\boldsymbol{\theta}$ for each task $\tau$ with stochastic gradient descent (SGD). Although MAML allows multiple-step being taken to update $\boldsymbol{\theta}$, we will consider the case only one step being taken for simplicity. The update equation is as follows:

$$\boldsymbol{\theta}_\tau' = \boldsymbol{\theta} - \alpha \nabla_{\boldsymbol{\theta}} L_\tau\left(\boldsymbol{\theta}\right), \tag{1}$$

where $\alpha$ is a step size referred to as the inner learning rate, $L_\tau\left(\boldsymbol{\theta}\right)$ is the loss of $\tau$, and $\nabla_{\boldsymbol{\theta}} L_\tau(\boldsymbol{\theta})$ is an estimate of the true gradient. The data used for this update is called training data. Next, MAML resamples data, referred to as test data, from each $\tau$ and computes the loss at the updated parameters $\boldsymbol{\theta}_\tau'$, obtaining $L_\tau(\boldsymbol{\theta}_\tau')$ for each task. Finally, to determine $\boldsymbol{\theta}$ that can be adapted to $\boldsymbol{\theta}_\tau'$ for all tasks, $\boldsymbol{\theta}$ is updated with the gradient of a sum of loss values $L_\tau(\boldsymbol{\theta}_\tau')$ over all tasks. In other words,

$$\boldsymbol{\theta} \leftarrow \boldsymbol{\theta} - \beta \nabla_{\boldsymbol{\theta}} \sum_{\tau \sim P(\tau)} L_\tau\left(\boldsymbol{\theta}_\tau'\right), \tag{2}$$

where $\beta$ is the learning rate called the meta-learning rate and $\nabla_{\boldsymbol{\theta}} \sum_{\tau \sim P(\tau)} L_\tau\left(\boldsymbol{\theta}_\tau'\right)$ is an estimate of the true gradient by using the test data. Though learning rates $\alpha$ and $\beta$ can be tuned during training or different for each task in practice, we will think them as fixed scalar hyperparameters.

### 2.2 NEGATIVE GRADIENT PENALTY

Unless otherwise noted, we will consider the case of only one step being made per update, and the data are not resampled to compute the loss for updating $\boldsymbol{\theta}$. The case of multiple steps and that of training data and test data being separated are considered in Appendix A. The gradient of the loss at $\boldsymbol{\theta}_\tau'$ is $\boldsymbol{g}_\tau(\boldsymbol{\theta}_\tau') = \nabla_{\boldsymbol{\theta}} L_\tau\left(\boldsymbol{\theta}_\tau'\right) = \nabla_{\boldsymbol{\theta}} \boldsymbol{\theta}_\tau' \frac{\partial L_\tau}{\partial \boldsymbol{\theta}_\tau'}$, where $\boldsymbol{g}(\cdot)$ is the gradient of $L(\cdot)$ with respect to $\boldsymbol{\theta}$. If $\alpha$ is small, we can assume that $\boldsymbol{I} \frac{\partial L_\tau}{\partial \boldsymbol{\theta}_\tau'} = \boldsymbol{g}_\tau(\boldsymbol{\theta})$; this seems to hold since $\alpha$ is usually small (Finn et al., 2017). Then,

$$\nabla_{\boldsymbol{\theta}} L_\tau\left(\boldsymbol{\theta}_\tau'\right) = \nabla_{\boldsymbol{\theta}} \boldsymbol{\theta}_\tau' \frac{\partial L_\tau}{\partial \boldsymbol{\theta}_\tau'} = \left(\boldsymbol{I} - \alpha \nabla_{\boldsymbol{\theta}}^2 L_\tau\right) \frac{\partial L_\tau}{\partial \boldsymbol{\theta}_\tau'} \tag{3}$$

$$\approx \boldsymbol{g}_\tau(\boldsymbol{\theta}) - \alpha \boldsymbol{H}_\tau(\boldsymbol{\theta}) \boldsymbol{g}_\tau(\boldsymbol{\theta}). \tag{4}$$

The result but not the procedure with the approximation is the same as that with the well-known first-order approximation as long as data are not resampled and only one step being taken. The first order approximation has been mentioned by Finn et al. (2017) and extensively studied by Nichol et al. (2018) and Fallah et al. (2019). It is known that the error induced by the first-order approximation does not degrade the performance so much in practice (Finn et al., 2017). Also, Fallah et al. (2019) theoretically proved that the first-order approximation of MAML does not affect convergence result when each task is similar to each other or $\alpha$ is small enough.

For simplicity, we will assume that only one task is considered during training, omitting task index $\tau$. Therefore, instead of $\sum_{\tau \sim P(\tau)} L_\tau\left(\boldsymbol{\theta}_\tau'\right)$, we will consider $L(\boldsymbol{\theta}')$ as the loss of the simplified MAML. Since the MAML loss is just a sum of task-specific loss, extension to the case of multiple tasks being considered is straightforward, as provided in Appendix A.1.1. Because $\nabla_{\boldsymbol{\theta}}(\boldsymbol{g}(\boldsymbol{\theta})^\top \boldsymbol{g}(\boldsymbol{\theta})) = 2\boldsymbol{H}(\boldsymbol{\theta})\boldsymbol{g}(\boldsymbol{\theta})$, if we define $\tilde{L}(\boldsymbol{\theta}) = L(\boldsymbol{\theta}')$,

$$\tilde{L}(\boldsymbol{\theta}) \approx L(\boldsymbol{\theta}) - \frac{\alpha}{2} \boldsymbol{g}(\theta)^\top \boldsymbol{g}(\boldsymbol{\theta}). \tag{5}$$

The above means that the simplified MAML can be regarded as optimization with the negative gradient penalty. We will analyze this simplified MAML loss in Section 3. It can also be interpreted as a Taylor series expansion of the simplified MAML loss for the first-order term, up to scale:

$$\tilde{L}(\boldsymbol{\theta}) = L(\boldsymbol{\theta} - \alpha\nabla_{\boldsymbol{\theta}}L(\boldsymbol{\theta})) \tag{6}$$
$$\approx L(\boldsymbol{\theta}) - \alpha\nabla_{\boldsymbol{\theta}}L(\boldsymbol{\theta})^{\top}\nabla_{\boldsymbol{\theta}}L(\boldsymbol{\theta}) \quad \text{(Taylor series expansion)}$$
$$= L(\boldsymbol{\theta}) - \alpha\boldsymbol{g}(\boldsymbol{\theta})^{\top}\boldsymbol{g}(\boldsymbol{\theta}). \tag{7}$$

The fact that the simplified MAML is optimization with the negative gradient penalty is worth keeping in mind. Because the goal of gradient-based optimization is to find a point where the gradient is zero, a bias that favors a larger gradient is highly likely to make training unstable; this can be a cause of instability of MAML (Antoniou et al., 2019). In fact, as shown in Fig. 1, the gradient norm becomes larger during training, as do the gradient inner products, as Guiroy et al. (2019) observed.

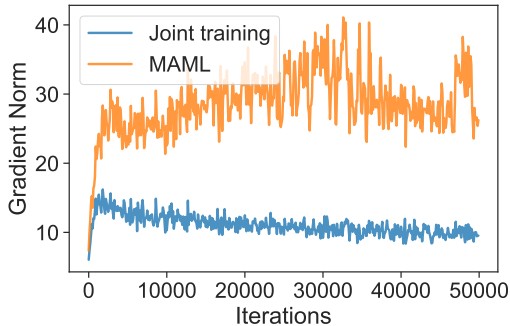

Figure 1. Gradient norm during training. We compute the norm per task and subsequently compute their average. *Joint training* shows when $\alpha = 0$, and *MAML* is when $\alpha = 1\text{e-}2$. These results are computed using training data, but those determined using test data behave similarly. The total number of iterations is 50000, $\beta = 1\text{e-}3$ and the Adam optimizer is used. Other settings are the same as those in Section 5.2.

## 3 LEARNING RATE FOR LOCAL CONVERGENCE

In this section, we will derive a sufficient condition of learning rate $\alpha$ and $\beta$ for local convergence from any point in the vicinity of the local minima. To this end, we will assume that only one step is taken for update and training data and test data are not distinguished as we did in Section 2. Also, we do not consider SGD but steepest GD to derive the condition. In other words, same training data are assumed to be used continuously for updating parameters during training. First, we will consider the case of only single task being considered. Next, we will consider the case of multiple tasks being considered.

### 3.1 SINGLE TASK

#### 3.1.1 CONDITION FOR INNER LEARNING RATE $\alpha$

First, we derive the sufficient condition of learning rate $\alpha$. To this end, we will consider the sufficient condition that a fixed point is a local minimum. Taking the Taylor series for the second-order term at a fixed point $\boldsymbol{\theta}^*$, the simplified MAML loss is

$$\tilde{L}(\boldsymbol{\theta}) \approx \tilde{L}(\boldsymbol{\theta}^*) + \frac{1}{2}(\boldsymbol{\theta} - \boldsymbol{\theta}^*)^{\top}\tilde{\boldsymbol{H}}(\boldsymbol{\theta} - \boldsymbol{\theta}^*). \tag{8}$$

where $\tilde{\boldsymbol{H}} = \boldsymbol{H} - \alpha\left(\boldsymbol{\mathsf{T}}\boldsymbol{g} + \boldsymbol{H}^2\right)$ is the Hessian matrix of $\tilde{L}(\boldsymbol{\theta})$ at $\boldsymbol{\theta}^*$ and $\boldsymbol{g} = \nabla_{\boldsymbol{\theta}}L(\boldsymbol{\theta}^*) \in \mathbb{R}^d$, $\boldsymbol{H} = \nabla_{\boldsymbol{\theta}}^2 L(\boldsymbol{\theta}^*) \in \mathbb{R}^{d \times d}$, and $\boldsymbol{\mathsf{T}} = \nabla_{\boldsymbol{\theta}}^3 L(\boldsymbol{\theta}^*) \in \mathbb{R}^{d \times d \times d}$. The calculation of $\tilde{\boldsymbol{H}}$ is presented in Appendix B. We calculated the magnitudes of $\boldsymbol{\mathsf{T}}\boldsymbol{g}$ and $\boldsymbol{H}^2$ numerically and observed that $\boldsymbol{\mathsf{T}}\boldsymbol{g}$ was much smaller than $\boldsymbol{H}^2$ in practice. Hence, we will ignore $\boldsymbol{\mathsf{T}}\boldsymbol{g}$ while deriving the condition and will

thus assume that $\tilde{\boldsymbol{H}} = \boldsymbol{H} - \alpha \boldsymbol{H}^2$. Further details are provided in Appendix C, and the case of $\mathbf{T}g$ being considered is provided in Appendix D. Since $\boldsymbol{P}\boldsymbol{\Lambda}_{\tilde{\boldsymbol{H}}}\boldsymbol{P}^\top = \boldsymbol{P}[\boldsymbol{\Lambda}_{\boldsymbol{H}} - \alpha \boldsymbol{\Lambda}_{\boldsymbol{H}}^2]\boldsymbol{P}^\top$ where $\boldsymbol{\Lambda}_{\tilde{\boldsymbol{H}}}$ is a diagonal matrix with entries that are eigenvalues of $\tilde{\boldsymbol{H}}$ and $\boldsymbol{P}$ is a matrix with rows that are eigenvectors of $\tilde{\boldsymbol{H}}$, the sufficinet condition of $\alpha$ for $\boldsymbol{\theta}^*$ to be a local minimum is

$$\forall i, \ \lambda(\tilde{\boldsymbol{H}})_i = \lambda(\boldsymbol{H})_i - \alpha\lambda(\boldsymbol{H})_i^2 > 0 \tag{9}$$

$$\Rightarrow \forall i, \ \alpha < \frac{1}{\lambda(\boldsymbol{H})_i}. \tag{10}$$

Note that $\lambda(\boldsymbol{A})_i$ represents the $i$th eigenvalue of matrix $\boldsymbol{A}$. Hence, the sufficient condition of $\alpha$ for $\boldsymbol{\theta}^*$ to be a local minimum is

$$\forall i, \ \alpha < \frac{1}{\lambda(\boldsymbol{H})_i}. \tag{11}$$

Therefore, $\alpha_c$ is the inverse of the largest eigenvalue of $\boldsymbol{H}$.

### 3.1.2 CONDITION FOR META-LEARNING RATE $\beta$

Next, we derive the sufficient condition of meta-learning rate $\beta$ for the simplified MAML to locally converge to the local minima discussed above from any point in the vicinity of the local minimum. This is an extension of research of LeCun et al. (1998). Since $\boldsymbol{P}\boldsymbol{P}^\top = \boldsymbol{I}$, the simplified MAML loss can be written as

$$\tilde{L}(\boldsymbol{\theta}) \approx \tilde{L}(\boldsymbol{\theta}^*) + \frac{1}{2}((\boldsymbol{\theta} - \boldsymbol{\theta}^*)^\top \boldsymbol{P})\boldsymbol{P}^\top \tilde{\boldsymbol{H}} \boldsymbol{P}(\boldsymbol{P}^\top(\boldsymbol{\theta} - \boldsymbol{\theta}^*)). \tag{12}$$

By using the simplified loss defined in Eq. 8, the update equation of the parameter $\boldsymbol{\theta}$ with GD is

$$\boldsymbol{\theta}(t+1) = \boldsymbol{\theta}(t) - \beta\nabla_{\boldsymbol{\theta}}\tilde{L}(\boldsymbol{\theta}) \tag{13}$$

$$= \boldsymbol{\theta}(t) - \beta\tilde{\boldsymbol{H}}(\boldsymbol{\theta} - \boldsymbol{\theta}^*) \tag{14}$$

where $t$ ($t = 0, ..., M$) is iteration and $M$ is the total number of iterations. Hence, $\boldsymbol{\theta}(t+1) - \boldsymbol{\theta}^* = (\boldsymbol{I} - \beta\tilde{\boldsymbol{H}})(\boldsymbol{\theta}(t) - \boldsymbol{\theta}^*)$. If we denote $\boldsymbol{P}^\top(\boldsymbol{\theta} - \boldsymbol{\theta}^*)$ by $\boldsymbol{v}$, the simplified MAML loss in Eq. 12 is $\tilde{L}(\boldsymbol{v}) \approx \tilde{L}(0) + \frac{1}{2}\boldsymbol{v}^\top\boldsymbol{\Lambda}_{\tilde{\boldsymbol{H}}}\boldsymbol{v}$. Because the gradient of $\tilde{L}(\boldsymbol{v})$ for $\boldsymbol{v}$ is $\nabla_{\boldsymbol{v}}\tilde{L}(\boldsymbol{v}) = \boldsymbol{\Lambda}_{\tilde{\boldsymbol{H}}}\boldsymbol{v}$, the update equation of $\boldsymbol{v}$ is

$$\boldsymbol{v}(t+1) = \boldsymbol{v}(t) - \beta\boldsymbol{\Lambda}_{\tilde{\boldsymbol{H}}}\boldsymbol{v}(t) = (\boldsymbol{I} - \beta\boldsymbol{\Lambda}_{\tilde{\boldsymbol{H}}})\boldsymbol{v}(t), \tag{15}$$

where $\boldsymbol{v}(t)$ is the value of $\boldsymbol{v}$ during iteration $t$. Assuming that Eq. 11 holds, the sufficinet condition of $\beta$ is as follows: for all $i$,

$$|1 - \beta\lambda(\boldsymbol{H} - \alpha\boldsymbol{H}^2)_i| = |1 - \beta(\lambda(\boldsymbol{H})_i - \alpha\lambda(\boldsymbol{H})_i^2)| < 1 \tag{16}$$

$$\Rightarrow -1 + \beta(\lambda(\boldsymbol{H})_i - \alpha\lambda(\boldsymbol{H})_i^2) < 1 \ \ (\because \lambda(\boldsymbol{H})_i - \alpha\lambda(\boldsymbol{H})_i^2 > 0 \text{ holds because of Eq. 11}) \tag{17}$$

$$\Rightarrow \beta < \frac{2}{\lambda(\boldsymbol{H})_i - \alpha\lambda(\boldsymbol{H})_i^2}. \tag{18}$$

Consequently, the sufficient condition for the simplified MAML to locally converge to local minima from any point in the vicinity of the local minima is as follows:

$$\forall i, \ \alpha < \frac{1}{\lambda(\boldsymbol{H})_i} \ \wedge \ \beta < \frac{2}{\lambda(\boldsymbol{H})_i - \alpha\lambda(\boldsymbol{H})_i^2}. \tag{19}$$

Vanilla GD with learning rate $\beta$ corresponds to MAML if $\alpha = 0$. In this case, $\beta < \frac{2}{\lambda_{max}}$ is the condition of $\beta$, where $\lambda_{max}$ is the largest eigenvalue of $\boldsymbol{H}$, because $2/\lambda_{max}$ is smaller than any other $2/\lambda_i$ (LeCun et al., 1998). Though this holds for the simplified MAML as well, this is not the case if $\alpha$ is close to $\alpha_c$. The reason is that $\beta_c$ diverges as $\alpha$ approaches $\frac{1}{\lambda(\boldsymbol{H})_i}$, or $\alpha_c$ as Eq. 18 indicates. Hence, for the simplified MAML we must consider not only the largest but also other eigenvalues and in particular, the second-largest eigenvalue. In short, unlike when vanilla GD is employed, $\beta_c$ depends on $\alpha$ in the case of the simplified MAML, and $\beta_c$ is expected to be larger if $\alpha$ is close to $\alpha_c$, as shown in Fig. 2. This finding is validated by experiments presented in Section 5.

## 3.2 MULTIPLE TASKS

We discussed the case of only one task being available during training in Section 3.1. In this section, we derive upper bounds of $\alpha$ and $\beta$ that apply if multiple tasks $\tau \sim P(\tau)$ are considered. Assumptions, except that multiple tasks are considered, are the same as those in Section 3.1.

### 3.2.1 CONDITION FOR $\alpha$

Now, we define the simplified meta-objective as a sum of task-specific objectives: $\hat{L}(\boldsymbol{\theta}) = \sum_{\tau \sim P(\tau)} \tilde{L}_\tau(\boldsymbol{\theta})$. Then, at a fixed point $\boldsymbol{\theta}^*$,

$$\hat{L}(\boldsymbol{\theta}) \approx \hat{L}(\boldsymbol{\theta}^*) + \frac{1}{2}(\boldsymbol{\theta} - \boldsymbol{\theta}^*)^\top \hat{\boldsymbol{H}}(\boldsymbol{\theta} - \boldsymbol{\theta}^*) \tag{20}$$

$$= \sum_{\tau \sim \boldsymbol{P}(\tau)} \left( L_\tau(\boldsymbol{\theta}^*) + \frac{1}{2}(\boldsymbol{\theta} - \boldsymbol{\theta}^*)(\boldsymbol{H}_\tau - \alpha \boldsymbol{H}_\tau^2)(\boldsymbol{\theta} - \boldsymbol{\theta}^*) \right) \tag{21}$$

$$= \sum_{\tau \sim \boldsymbol{P}(\tau)} \left( L_\tau(\boldsymbol{\theta}^*) + \frac{1}{2}(\boldsymbol{\theta} - \boldsymbol{\theta}^*)^\top \boldsymbol{P}_\tau (\boldsymbol{\Lambda}_{\boldsymbol{H}_\tau} - \alpha \boldsymbol{\Lambda}_{\boldsymbol{H}_\tau}^2) \boldsymbol{P}_\tau^\top (\boldsymbol{\theta} - \boldsymbol{\theta}^*) \right), \tag{22}$$

where $\hat{\boldsymbol{H}}$ is the Hessian matrix of $\hat{L}(\boldsymbol{\theta})$ at $\boldsymbol{\theta}$. Note that we ignore $\boldsymbol{\mathsf{T}}_\tau \boldsymbol{g}_\tau$ as we did in Section 3.1.1. Since $\boldsymbol{H}_\tau - \alpha \boldsymbol{H}_\tau^2$ is not necessarily simultaneously diagonalizable for each task $\tau$, we cannot exactly express eigenvalues of $\hat{\boldsymbol{H}}$ as function of $\lambda(\boldsymbol{H}_\tau)$ and $\alpha$. Therefore, instead of the exact value of $\alpha_c$, we will derive an upper bound of $\alpha_c$.

If $\boldsymbol{\theta}^*$ is a local minimum for all $\tilde{L}_\tau(\boldsymbol{\theta}^*)$, $\boldsymbol{\theta}^*$ is a local minimum for $\hat{L}(\boldsymbol{\theta})$ as well. Hence, if all eigenvalues of $\tilde{\boldsymbol{H}}_\tau = \boldsymbol{H}_\tau - 2\alpha \boldsymbol{H}_\tau^2$ are positive for all tasks, those of $\hat{\boldsymbol{H}}$ are positive as well. Therefore, if the inequality

$$\forall \tau, i, \quad \alpha < \frac{1}{\lambda(\boldsymbol{H}_\tau)_i} \tag{23}$$

holds, it guarantees that the condition that $\boldsymbol{\theta}^*$ is a local minimum of $\hat{L}(\boldsymbol{\theta})$ is satisfied. Note that this is a sufficient condition if $\boldsymbol{\theta}^*$ is a local minimum for all $\tilde{L}_\tau(\boldsymbol{\theta}^*)$. Since $\boldsymbol{\theta}^*$ can be a local minimum of $\hat{L}(\boldsymbol{\theta})$ even when it is not a local minimum for all $\tilde{L}_\tau(\boldsymbol{\theta}^*)$, the upper bound of $\alpha$ seems to be larger than that in Eq. 23 in practice.

### 3.2.2 CONDITION FOR $\beta$

The analysis for $\beta$ is also basically the same as that performed in Section 3.1.2. Denoting $\boldsymbol{P}_\tau^\top (\boldsymbol{\theta} - \boldsymbol{\theta}^*)$ by $\boldsymbol{v}_\tau$, we obtain

$$\sum_\tau \boldsymbol{v}_\tau(t+1) = \sum_\tau \left( \boldsymbol{I} - \beta \left[ \boldsymbol{\Lambda}_{\boldsymbol{H}_\tau} - \alpha \boldsymbol{\Lambda}_{\boldsymbol{H}_\tau}^2 \right] \right) \boldsymbol{v}_\tau(t). \tag{24}$$

Since $\tilde{\boldsymbol{H}}_\tau$ is not always simultaneously diagonalizable for each task, as mentioned in Section 3.2.1, $\boldsymbol{P}_\tau$ differs from task to task in general. Hence, we will consider an upper bound of $\beta_c$ as we did for $\alpha_c$ in Section 3.2.1. Accordingly, if both Eq. 23 and

$$|1 - \beta(\lambda(\boldsymbol{H}_\tau)_i - \alpha\lambda(\boldsymbol{H}_\tau)_i^2)| < 1 \tag{25}$$

hold for any eigenvalue $\lambda_i$ of any task $\tau$, it guarantees that $\beta$ satisfies the condition for local convergence. Therefore, a sufficient for the simplified MAML to locally converge to local minima from any point in the neighborhood of the local minima in the case of multiple tasks is as follows:

$$\forall \tau, i, \quad \alpha < \frac{1}{\lambda(\boldsymbol{H}_\tau)_i} \quad \wedge \quad \beta < \frac{2}{\lambda(\boldsymbol{H}_\tau)_i - \alpha\lambda(\boldsymbol{H}_\tau)_i^2} \tag{26}$$

## 4 RELATED WORKS

Sevral papers have investigated MAML and proposed various algorithms (Nichol et al., 2018; Guiroy et al., 2019; Eshratifar et al., 2018; Antoniou et al., 2019; Fallah et al., 2019; Khodak et al.,

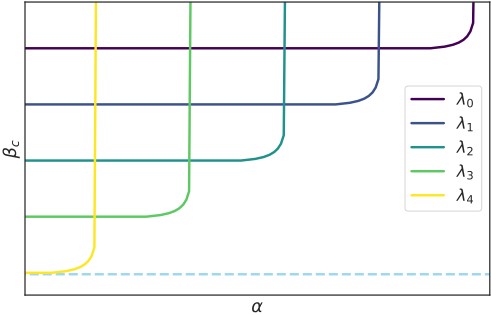

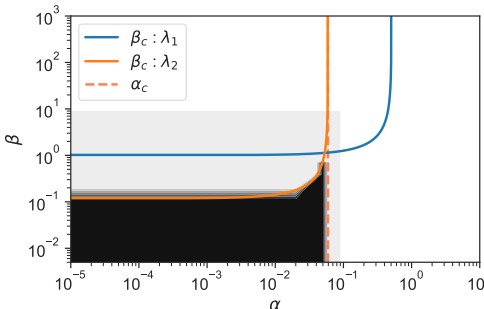

Figure 2. Curves of $\beta_c$ as a function of $\alpha$ for eigenvalues of the Hessian, $\lambda_0 < ... < \lambda_4$. Parameter $\beta$ is supposed to be smaller than $\beta_c$ for both $\lambda_4$ and $\lambda_3$. Hence, $\beta$ should be chosen from the colored area. Since $\alpha$ must satisfy $\alpha < \frac{1}{\lambda_i}$, $\alpha$ should also be in the colored region. The dashed line shows $\beta_c$ if $\alpha = 0$. If $\alpha \approx \alpha_c$, $\beta_c$ is larger than that at $\alpha = 0$.

Figure 3. Training loss of linear regression. The area colored in black is when the loss is below 1e-2, and that in gray is when the loss is over 1e-2. Uncolored region is not considered. $\beta_c : \lambda_i$ shows $\beta_c$ of $\lambda_i$, where $\lambda_1 < \lambda_2$. The dashed line is $\alpha_c$. Theoretical $\beta_c$ and $\alpha_c$ correspond to empirical ones.

2019; Vuorio et al., 2018; Finn et al., 2019; Deleu & Bengio, 2018; Liu & Theodorou, 2019; Deleu & Bengio, 2018; Grant et al., 2018). Nichol et al. (2018) studied the first-order MAML family in detail and showed that the MAML gradient could be decomposed into two terms: a term related to joint training and a term responsible for increasing the inner product between gradients for different tasks.

Guiroy et al. (2019) investigated the generalization ability of MAML. The researchers observed that generalization was correlated with the average gradient inner product and that flatness of the loss surface, often thought to be an indicator of strong generalizability in normal neural network training, was not necessarily related to generalizability in the case of MAML. Eshratifar et al. (2018) also noted that the average gradient inner product was important. Hence, the authors proposed an algorithm that considered the relative importance of each parameter based on the magnitude of the inner product between the task-specific gradient and the average gradient. Although the above studies were cognizant of the importance of the inner product of the gradients, they did not explicitly insert the negative gradient inner product, which is the negative squared gradient norm with simplifications, as a regularization term. To consider the simplified MAML as optimization with a regularization term is a contribution of our study.

Antoniou et al. (2019) enumerated five factors that could cause training MAML to be difficult. Then, they authors proposed an algorithm to address all of these problems and make training MAML easier and more stable. Behl et al. (2019), like us, pointed out that tuning the inner learning rate $\alpha$ and meta-learning rate $\beta$ was troublesome. The authors approached this problem by proposing an algorithm that tuned learning rates automatically during training.

Fallah et al. (2019) studied the convergence theory of MAML. They proposed a method for selecting meta-learning rate by approximating smoothness of the loss. Based on this result, they proved that MAML can find an $\varepsilon$-first-order stationary point after sufficient number of iterations. On the other hand, we studied the relationship between the sufficient conditions of inner learning rate $\alpha$ and meta-learning rate $\beta$ and showed that how the largest possible $\beta$ is affected by the value of $\alpha$.

## 5 EXPERIMENTS

In this section, we will present the results of experiments to confirm our expectation that MAML allows larger $\beta$ if $\alpha$ is close to its upper bound. First, we will show the result of linear regression with simplifications used in Section 3.1. Because linear regression is convex optimization, the result is expected to exactly match the theory presented in Section 3.1. Second, to check if our expectation is confirmed in practice as well, we will present results of the practical case without any simplification. In particular, we conducted sinusoid regression and classification of Omniglot and MiniImagenet

datasets with a multilayer perceptron and a convolutional neural network (CNN). Note that the meta-objective used for experiments is not a sum of task-specific objectives but a mean of them.

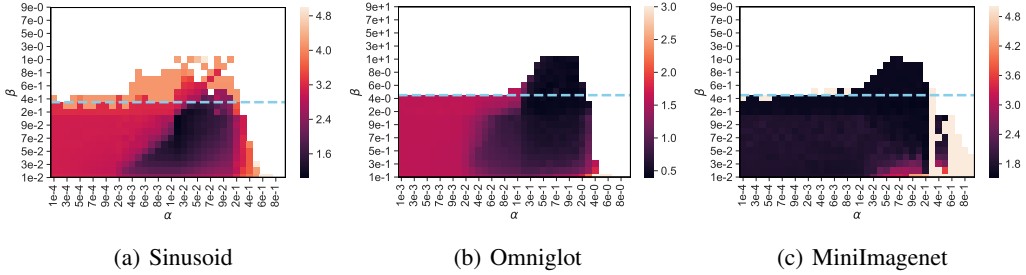

(a) Sinusoid          (b) Omniglot          (c) MiniImagenet

Figure 4. Training losses for (a) sinusoid regression, (b) Omniglot classification, and (c) MiniImagenet classification at various values of $\alpha$ and $\beta$ after a fixed number of iterations. The area with no color represents the diverged losses, and the dashed line indicates the values of $\beta$ above which the loss diverges for $\alpha = 0$. The maximum possible $\beta$ is larger if $\alpha$ is close to the value above which the losses diverge than that at $\alpha = 0$.

## 5.1 LINEAR REGRESSION

We performed a linear regression, where the task is to regress a linear function with scale parameter in the range of [0, 5.0] and bias parameter in the range of [0, 5.0] based on data points in the range of [-5.0, 5.0]. The true function has the same architecture as that of the model. We employed the steepest gradient descent method to minimize the mean squared loss, where 1 step was taken during update. Only one task was considered during training and the same data was used to update the task-specific parameter and the meta parameter as we did in Section 3.1. Using these settings, we computed the training loss after 500 iterations with $\alpha$ in the range of [1e-5, 9e-2] and $\beta$ in the range of [5e-3, 9e+0]. The eigenvalues are those of the Hessian matrix of the training loss at the end of the training, where $\alpha = 5e-2$ and $\beta = 7e-1$. We chose this training loss because it was thought to be the closest to minima. Fig. 3 shows the training losses at various values of $\alpha$ and $\beta$. Horizontal axis indicates $\alpha$ and vertical axis indicates $\beta$. The curves are $\beta_c$ of two eigenvalues and the dashed line shows $\alpha_c$. In the case of linear regression with simplifications used in Section 3.1, the result of numerical experiment shows good agreement with upper bounds of $\beta_c$ and $\alpha_c$ that we derived in Section 3.1, as shown in Fig. 3.

## 5.2 SINUSOID REGRESSION

We conducted a sinusoid regression, where each task is to regress a sine wave with amplitude in the range of [0.1, 5.0] and phase in the range of [0, $\pi$] based on data points in the range of [-5.0, 5.0]. A multilayer perceptron with two hidden units of size 40 and ReLU was trained with SGD. The batch size of data was 10, the number of tasks was 100, and 1 step was taken for update. Using these settings, we computed the training loss after 500 iterations with $\alpha$ in the range of [1e-4, 9e-1] and $\beta$ in the range of [1e-2, 9e-0]. Fig. 4 (a) shows the training losses with various values of $\alpha$ and $\beta$. The dashed line indicates $\beta$ of $\alpha = 0$ over which training loss diverges. According to Fig. 4 (a), if $\alpha$ is close to the value above which the losses diverge, a larger $\beta$ can be used. As explained above, we did not put any simplification as we did in Section 3 and 5.1 in the experiment, meaning that we used different data for updating task-specific parameter and meta parameter, considered multiple tasks, and employed not steepest GD but SGD as optimizer. Despite simplifications, surprisingly, this result confirms the expectation that MAML allows larger $\beta$ if $\alpha$ is close to $\alpha_c$.

## 5.3 CLASSIFICATION

We performed classification of the Omniglot and MiniImagenet datasets (Lake et al., 2011; ravi & Larochelle, 2017), which are benchmark datasets for few-shot learning. The model used was essentially the same as that Finn et al. (2017), and hence, Vinyals et al. (2016) used. The task is a five-way one-shot classification, where the query size is 15, the number of update steps is two, and the task batch size is 32 for Omniglot and four for MiniImagenet. In this setup, we computed

the training losses after 100 iterations for the Omniglot dataset and one epoch for the MiniImagenet dataset with various values of $\alpha$ and $\beta$; for Omniglot, $\alpha$ was in the range of [1e-3, 9e-0] and $\beta$ was in the range of [1e-1, 9e+1], and for MiniImagenet, $\alpha$ was in the range of [1e-4, 9e-1] and $\beta$ was in the range of [1e-2, 9e-0]. Fig. 4 (b) and (c) show the training losses of classification task at various values of $\alpha$ and $\beta$. The dashed line indicates $\beta$ of $\alpha = 0$ above which training loss diverges. As shown in Fig. 4 (b) and (c), the maximum $\beta$ is larger at large $\alpha$. Even though the model architecture is composed of convolutional layer, max-pooling, and batch normalization (Ioffe & Szegedy, 2015) and practical dataset is used for training, our expectation is confirmed in this case as well. This result confirms that our theory is applicable in practice.

Our experimental result confirms that larger $\alpha$ is good for stabilizing MAML training. According to Fig. 4 (a), (b) and (c), moreover, while large $\beta$ does not necessarily make training loss smaller, employing large $\alpha$ leads to smaller training loss comparatively. These result has a practical implication for tuning the learning rates: first, the largest possible $\alpha$ should be identified, and $\beta$ may be subsequently tuned based on the value of $\alpha$. Once you identify $\alpha_c$, MAML is likely to work well even if meta-learning rate is roughly chosen. Taking large $\alpha$ is desirable for the goal of MAML as well. The aim of MAML is to find a good initial parameter which quickly adapt to new tasks and the quickness is determined by inner learning rate $\alpha$ because $\alpha$ determines the step size from initial parameter to task-specific parameter when model is fine-tuned. Therefore, identifying $\alpha_c$ is not only good for robustifying the model against divergence but also good for finding good initial parameter.

## 6 CONCLUSIONS

We regard a simplified MAML as training with the negative gradient penalty. Based on this formulation, we derived the sufficient condition of the inner learning rate $\alpha$ and the meta-learning rate $\beta$ for the simplified MAML to locally converge to local minima from any point in the vicinity of the local minima. We showed that the upper bound of $\beta$ required for the simplified MAML to locally converge to local minima depends on $\alpha$. Moreover, we found that if $\alpha$ is close to its upper bound $\alpha_c$, the maximum possible meta-learning rate $\beta_c$ is larger than that used while training with ordinary SGD. This finding is validated by experiments, confirming that our theory is applicable in practice. According to this result, we propose a guideline for determining $\alpha$ and $\beta$; first, search for $\alpha$ close to $\alpha_c$; next, tune $\beta$ based on the selected value of $\alpha$.

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

## A    NEGATIVE GRADIENT PENALTY: GENERAL CASE

In Section 2, we explained the simplest case of training data and test data being the same, only one task being considered and only one step being taken for update. In this section, we analyze the cases of waiving one of these simplifications. Here, we will interpret the simplified MAML loss as a Taylor series expansion of the MAML loss for the first-order term.

### A.1    WHEN THE TRAINING DATA AND TEST DATA ARE DIFFERENT

When using different data to compute the losses for updating the task-specific parameters and the meta parameters as done in practical applications, the simplified loss is as follows:

$$\tilde{L}(\boldsymbol{\theta}) = L^{test}(\boldsymbol{\theta} - \alpha\nabla_{\boldsymbol{\theta}}L^{train}(\boldsymbol{\theta})) \tag{27}$$

$$\approx L^{test}(\boldsymbol{\theta}) - \alpha\boldsymbol{g}(\boldsymbol{\theta})^{test\top}\boldsymbol{g}(\boldsymbol{\theta})^{train}. \tag{28}$$

where $L^{train}$ and $L^{test}$ represent the loss values for training and test data, respectively, and $g^{train}$ and $g^{test}$ are gradients of these losses.

#### A.1.1    WHEN MULTIPLE TASKS ARE CONSIDERED

When multiple tasks are considered, as MAML does, the meta-loss $\hat{L}(\boldsymbol{\theta})$ is just a sum of losses $\tilde{L}_\tau(\boldsymbol{\theta})$ for all tasks $\tau$:

$$\hat{L}(\boldsymbol{\theta}) = \sum_{\tau\sim P(\tau)} \tilde{L}_\tau(\boldsymbol{\theta}) = \sum_{\tau\sim P(\tau)} \left(L_\tau^{test}(\boldsymbol{\theta}) - \alpha\boldsymbol{g}_\tau(\boldsymbol{\theta})^{test\top}\boldsymbol{g}_\tau(\boldsymbol{\theta})^{train}\right). \tag{29}$$

### A.2    WHEN $k$ STEPS ARE TAKEN DURING THE UPDATE

If task-specific parameters are updated with k-step SGD, the loss can be written as follows;

$$\tilde{L}_k(\boldsymbol{\theta}) = L_k(\boldsymbol{\theta}) - \alpha\left(\sum_i^k \boldsymbol{g}_k^\top(\boldsymbol{\theta})\boldsymbol{g}_i(\boldsymbol{\theta})\right). \tag{30}$$

Note that $L_i(\boldsymbol{\theta})$ is the loss computed with the data at the $i$th step, and $\boldsymbol{g}_i(\boldsymbol{\theta})$ is the gradient of the loss.

## B    CALCULATION OF $\tilde{H}$

Because $\tilde{H}$ is the Hessian matrix of $\tilde{L}$ at $\boldsymbol{\theta}^*$, we derive the Hessian of Eq. 8. Then,

$$\tilde{H} = \nabla_{\boldsymbol{\theta}}^2\tilde{L}(\boldsymbol{\theta}) \tag{31}$$

$$= \nabla_{\boldsymbol{\theta}}^2\left(L(\boldsymbol{\theta}) - \frac{\alpha}{2}\boldsymbol{g}(\theta)^\top\boldsymbol{g}(\boldsymbol{\theta})\right) \tag{32}$$

$$= \boldsymbol{H}(\boldsymbol{\theta}) - \alpha\nabla_{\boldsymbol{\theta}}\left(\boldsymbol{H}(\boldsymbol{\theta})\boldsymbol{g}(\boldsymbol{\theta})\right) \tag{33}$$

$$= \boldsymbol{H}(\boldsymbol{\theta}) - \alpha\left(\nabla_{\boldsymbol{\theta}}\boldsymbol{H}(\boldsymbol{\theta})\boldsymbol{g}(\boldsymbol{\theta}) + \boldsymbol{H}(\boldsymbol{\theta})\boldsymbol{H}(\boldsymbol{\theta})\right) \tag{34}$$

$$= \boldsymbol{H} - \alpha(\mathbf{T}\boldsymbol{g} + \boldsymbol{H}^2). \tag{35}$$

## C    MAGNITUDE OF $Tg$ AND $H^2$

We conducted a sinusoid regression with essentially the same condition that we explain in Section 5.2 except that the total number of iterations is 50000 and learnig rates are fixed. Parameters $\alpha$ and $\beta$ are 1e-2 and 1e-3 respectively. We calculated $\mathbf{T}\boldsymbol{g}$ numerically with the training error at the end of the training. As we showed in Section 3, especially large eigenvalues of $\tilde{H}$ are important for the upper bounds of learning rates. Therefore, if $\lambda(\mathbf{T}\boldsymbol{g} + \boldsymbol{H}^2)_{max} \approx \lambda(\boldsymbol{H}^2)_{max}$, we can ignore $\mathbf{T}\boldsymbol{g}$ when deriving the condition. We calculate the maximum and the second-largest eigenvalues of $\mathbf{T}\boldsymbol{g}$, $\boldsymbol{H}^2$ and $\mathbf{T}\boldsymbol{g} + \boldsymbol{H}^2$ of the trained model. As shown in Fig. 5 (a), $\lambda(\mathbf{T}\boldsymbol{g} + \boldsymbol{H}^2)_{max}$ is almost equal to

$\lambda(\boldsymbol{H}^2)_{max}$, and $\lambda(\mathsf{T}\boldsymbol{g})_{max}$ is by far smaller than them. Therefore, ignoring $\lambda(\mathsf{T}\boldsymbol{g})_{max}$ is reasonable when the conditions are derived. Furthermore, we calculate the Frobenius norm of $\mathsf{T}\boldsymbol{g}$ and $\boldsymbol{H}^2$. As Fig. 5 (b) indicates, the Frobenius norm of $\mathsf{T}\boldsymbol{g}$ is much smaller than that of $\boldsymbol{H}^2$, meaning that $\mathsf{T}\boldsymbol{g}$ is negligible in the sense of the magnitude of the norm as well. These results confirm that we can neglect $\mathsf{T}\boldsymbol{g}$ when considering $\tilde{\boldsymbol{H}}$.

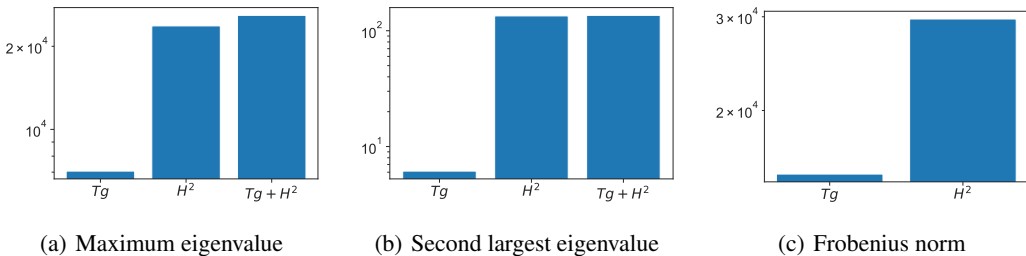

(a) Maximum eigenvalue    (b) Second largest eigenvalue    (c) Frobenius norm

Figure 5. (a): The maximum eigenvalues of $\mathsf{T}\boldsymbol{g}$, $\boldsymbol{H}^2$ and $\mathsf{T}\boldsymbol{g} + \boldsymbol{H}^2$. It is clear that the maximum eigenvalue of $\mathsf{T}\boldsymbol{g} + \boldsymbol{H}^2$ is almost the same as that of $\boldsymbol{H}^2$, while that of $\mathsf{T}\boldsymbol{g}$ is much smaller than them. (b): The second-largest eigenvalues of $\mathsf{T}\boldsymbol{g}$, $\boldsymbol{H}^2$ and $\mathsf{T}\boldsymbol{g} + \boldsymbol{H}^2$. Like (a), the second-largest eigenvalue of $\mathsf{T}\boldsymbol{g} + \boldsymbol{H}^2$ is almost equal to that of $\boldsymbol{H}^2$. (c): The Frobenius norm of $\mathsf{T}\boldsymbol{g}$ and $\boldsymbol{H}^2$. The Frobenius norm of $\mathsf{T}\boldsymbol{g}$ is much smaller than that of $\boldsymbol{H}^2$.

## D   CONVERGENCE CONDITION WHEN $Tg$ IS CONSIDERED

We assumed that $\mathsf{T}\boldsymbol{g}$ was negligible in Section 3. In this section, we derive a sufficient condition of $\alpha$ and $\beta$ for the simplified MAML to locally converge to local minima from any point in the vicinity of the local minima under some assumptions when $\mathsf{T}\boldsymbol{g}$ is considered.

### D.1   CONDITION FOR INNER LEARNING RATE $\alpha$ TO SATISFY

When $\mathsf{T}\boldsymbol{g}$ is considered, the Hessian matrix of the simplified MAML loss $\tilde{L}$ is $\tilde{\boldsymbol{H}} = \boldsymbol{H} - \alpha\left(\mathsf{T}\boldsymbol{g} + \boldsymbol{H}^2\right)$. Because $\boldsymbol{H} - \alpha\left(\mathsf{T}\boldsymbol{g} + \boldsymbol{H}^2\right)$ is a real symmetric matrix, it can be diagonalized. Then, the sufficient condition that a fixed point $\boldsymbol{\theta}^*$ is a local minimum is

$$\forall i, \ \ \lambda(\tilde{\boldsymbol{H}})_i = \lambda(\boldsymbol{H} - \alpha\left(\mathsf{T}\boldsymbol{g} + \boldsymbol{H}^2\right))_i > 0. \tag{36}$$

Note that $\mathsf{T}\boldsymbol{g}$ and $\boldsymbol{H}$ are not simultaneously diagonalizable, so we cannot decompose $\lambda(\tilde{\boldsymbol{H}})_i$ into eigenvalues of each matrix as we did in Section 3. Therefore, we have to consider the relationship among $\lambda(\tilde{\boldsymbol{H}})_i$, $\lambda(\boldsymbol{H})_i$ and $\lambda(\mathsf{T}\boldsymbol{g})_i$. In general, it is known that $n \times n$ Hermitian matrices $\boldsymbol{A}$ and $\boldsymbol{B}$ satisfy the following equation (Bhatia, 2001):

$$\boldsymbol{\lambda}^{\downarrow}(\boldsymbol{A}) + \boldsymbol{\lambda}^{\uparrow}(\boldsymbol{B}) \prec \boldsymbol{\lambda}(\boldsymbol{A} + \boldsymbol{B}), \tag{37}$$

where $\boldsymbol{\lambda}(\boldsymbol{A})$ represents a vector with elements that are eigenvalues of $\boldsymbol{A}$, $\uparrow$ indicates the operation of sorting a vector in the ascending order, and $\downarrow$ indicates that in the descending order. If two real vectors $\boldsymbol{x}, \boldsymbol{y} \in \mathbb{R}^d$ are related in the following way, $\boldsymbol{x}$ is said to be majorized by $\boldsymbol{y}$ and the relationship is written as $\boldsymbol{x} \prec \boldsymbol{y}$:

$$\sum_{i=1}^{k} x_i^{\downarrow} \leq \sum_{i=1}^{k} y_j^{\downarrow} \ \ \text{for} \ \ 1 \leq k \leq d, \tag{38}$$

$$\sum_{i=1}^{d} x_i^{\downarrow} = \sum_{i=1}^{d} y_i^{\downarrow}. \tag{39}$$

We define $\boldsymbol{A} = \boldsymbol{H} - \alpha\boldsymbol{H}^2$ and $\boldsymbol{B} = -\alpha\mathbf{T}\boldsymbol{g}$; then,

$$\boldsymbol{\lambda}^{\downarrow}(\boldsymbol{A}) + \boldsymbol{\lambda}^{\uparrow}(\boldsymbol{B}) \prec \boldsymbol{\lambda}(\boldsymbol{A} + \boldsymbol{B}) \tag{40}$$

$$\Rightarrow \boldsymbol{\lambda}^{\downarrow}(\boldsymbol{A}) - \alpha\boldsymbol{\lambda}^{\downarrow}(\mathbf{T}\boldsymbol{g}) \prec \boldsymbol{\lambda}(\boldsymbol{A} + \boldsymbol{B}) \tag{41}$$

$$\Rightarrow \sum_{i=1}^{k}(\lambda^{\downarrow}(\boldsymbol{A})_i - \alpha\lambda^{\downarrow}(\mathbf{T}\boldsymbol{g})_i) \leq \sum_{i=1}^{k}\lambda^{\downarrow}(\boldsymbol{A} + \boldsymbol{B})_i \quad \text{for} \quad 1 \leq k \leq d. \tag{42}$$

Let us suppose that for top $k$ eigenvalues of $\boldsymbol{A}$, $\mathbf{T}\boldsymbol{g}$, and $\boldsymbol{A} + \boldsymbol{B}$, conditions $\lambda(\boldsymbol{A} + \boldsymbol{B})_i \approx c_{\boldsymbol{A}+\boldsymbol{B}}$, $\lambda(\mathbf{T}\boldsymbol{g})_i \approx c_{\mathbf{T}\boldsymbol{g}}$ and $\lambda(\boldsymbol{A})_i \approx c_{\boldsymbol{A}}$ hold, where $c_{\boldsymbol{A}+\boldsymbol{B}}$, $c_{\boldsymbol{A}}$ and $c_{\mathbf{T}\boldsymbol{g}}$ are some constant values. Then,

$$\lambda(\boldsymbol{A})_i - \alpha\lambda(\mathbf{T}\boldsymbol{g})_i \leq \lambda(\boldsymbol{A} + \boldsymbol{B})_i \quad \text{for} \quad 1 \leq i \leq k \tag{43}$$

$$\Rightarrow \lambda(\boldsymbol{H} - \alpha\boldsymbol{H}^2)_i - \alpha\lambda(\mathbf{T}\boldsymbol{g})_i \leq \lambda(\boldsymbol{H} - \alpha(\mathbf{T}\boldsymbol{g} + \boldsymbol{H}^2))_i \quad \text{for} \quad 1 \leq i \leq k. \tag{44}$$

The sufficient condition for a fixed point $\boldsymbol{\theta}^*$ to be a local minimum is

$$\forall i, \ \lambda(\boldsymbol{H} - \alpha\boldsymbol{H}^2)_i - \alpha\lambda(\mathbf{T}\boldsymbol{g})_i > 0 \tag{45}$$

Now, we assume that all eigenvalues of $\boldsymbol{H}$ below a threshold are 0; $\lambda(\boldsymbol{H})_i \approx 0$ for $k < i \leq n$. In fact, many eigenvalues of the loss computed by a trained model such as a deep neural network are known to be very small. Thus, if the following condition is satisfied, it is enough to say that the condition is satisfied:

$$\begin{cases} \lambda(\boldsymbol{H} - \alpha\boldsymbol{H}^2)_i - \alpha\lambda(\mathbf{T}\boldsymbol{g})_i > 0 & \text{for} \quad 1 \leq i \leq k \\ \alpha\lambda(\mathbf{T}\boldsymbol{g})_i < 0 & \text{for} \quad k \leq i \leq d \end{cases} \tag{46}$$

$$\Rightarrow \begin{cases} \lambda(\boldsymbol{H})_i - \alpha\lambda(\boldsymbol{H})_{min}^2 - \alpha\lambda(\mathbf{T}\boldsymbol{g})_i > 0 & \text{for} \quad 1 \leq i \leq k \\ \lambda(\mathbf{T}\boldsymbol{g})_i < 0 & \text{for} \quad k \leq i \leq d \end{cases} \tag{47}$$

$$\Leftrightarrow \begin{cases} \lambda(\boldsymbol{H})_i - \alpha(\lambda(\boldsymbol{H})_{min}^2 + \lambda(\mathbf{T}\boldsymbol{g})_i) > 0 & \text{for} \quad 1 \leq i \leq k \\ \lambda(\mathbf{T}\boldsymbol{g})_i < 0 & \text{for} \quad k \leq i \leq d \end{cases} \tag{48}$$

$$\Rightarrow \begin{cases} \lambda(\boldsymbol{H})_i - \alpha\lambda(\mathbf{T}\boldsymbol{g})_i > 0 & \text{for} \quad 1 \leq i \leq k \\ \lambda(\mathbf{T}\boldsymbol{g})_i < 0 & \text{for} \quad k \leq i \leq d \end{cases} (\because \lambda(\boldsymbol{H})_{min} = 0) \tag{49}$$

In other words, if the smaller $d - k$ eigenvalues of $\mathbf{T}\boldsymbol{g}$ are not negative values, the fixed point is not guaranteed to be a minimum, and if they are negative, satisfying Eq. 49 is sufficient.

## D.2 Condition for meta learning rate $\beta$ to satisfy

Next, we derive a sufficient condition of meta-learning rate $\beta$. Our analysis will be based on the assumptions identical to those mentioned above. If the inequalities in Eq. 49 hold, it is sufficient for $\beta$ to satisfy

$$\forall i, \ -1 + \beta\left(\lambda(\boldsymbol{H} - \alpha\lambda\boldsymbol{H}^2)_i - \alpha\lambda(\mathbf{T}\boldsymbol{g})_i\right) \leq -1 + \beta\lambda(\boldsymbol{H} - \alpha(\mathbf{T}\boldsymbol{g} + \boldsymbol{H}^2))_i < 1 \tag{50}$$

$$\Rightarrow \forall i, \ \beta < \frac{2}{\lambda(\boldsymbol{H})_i - \alpha\lambda(\mathbf{T}\boldsymbol{g})_i} \ (\because \lambda(\boldsymbol{H})_{min} = 0) \tag{51}$$

for the condition of local convergence.

## E Experiment with Adam

We conducted the same experiment as that explained in Section 5.2 and 5.3, but this time, we used Adam optimizer instead of SGD for the meta-optimizer (Kingma & Ba, 2014). The results are different from those we showed in Section 5. For training with Adam, MAML no longer allows us to use larger $\beta_c$. Because Adam has more parameters that we have to consider than SGD does, this result is not at all surprising. Nonetheless, it is important to keep these facts in mind when Adam is used for MAML optimization.

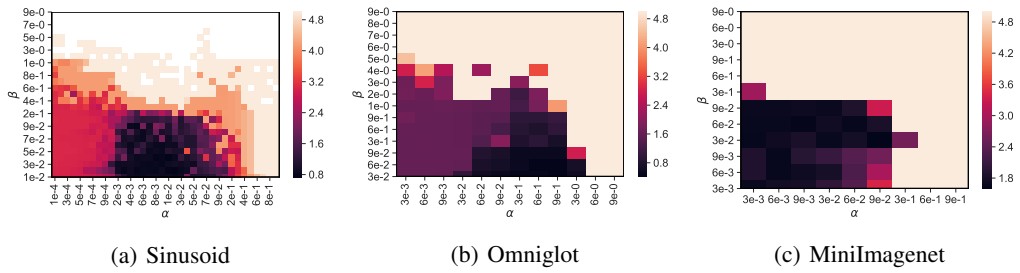

(a) Sinusoid  (b) Omniglot  (c) MiniImagenet

Figure 6. Training losses for (a) the sinusoid regression, (b) Omniglot classification and (c) MiniImagenet classification for various values of the inner learning rate $\alpha$ and meta-learning rate $\beta$ after a fixed number of iterations. The area with no color represents the diverged losses.

