# OpenReview forum: "Role of two learning rates in convergence of model-agnostic meta-learning"
_ICLR.cc/2020/Conference — Reject_

### Official Review · AnonReviewer1 · 2019-10-17
**Official Blind Review #1**

**Rating:** 3

**Review:**

In this paper, the authors tackled one important research problem in MAML, i.e., the optimization instability, by investigating the two learning rates. Though I appreciate the theoretic contribution this work makes, I am not sure with the practical significance of it.  Below please find my detailed comments.

Pros:
-	In this work, the authors focused on an important problem – training MAML is kind of unstable and tricky, so that developing guidelines that stabilize MAML or its first-order approximations is of significance.
-	This work theoretically discusses the relationship between the inner loop learning rate and the outer one, under a set of assumptions.
-	The paper is well written and easy to follow.

Cons:
-	Some of the simplifications for proving are empirical, so that the proof itself is not that rigorous.
  o	For example, In Section 3.1.1, the authors ignored Tg based on an observation that Tg is small. However, even in the Appendix, the authors did not explicate the experimental setting where they reach such a conclusion. Will Tg be always small then, in any task and any dataset?
-	The take-away of this work is not clear, in other words, it makes small contribution to the practical training of MAML.
  o	Practically, we often train MAML in 5 or 10 steps instead of only one step. Will the conclusion apply to such setups? Or will the model itself diverge during the inner loop as more steps are taken, provided with a larger value of \alpha?
  o	Practically, we merely use vanilla SGD. The figure in the Appendix shows that the main conclusion actually does not apply to Adam, which disempower the practicability of the theoretical guideline.
  o	How can we search the “largest possible” inner loop learning rate? Should we still use grid-search or heuristic search? In that case, what is the implication/shortcut that this proposed guideline brings? We still reach a stable training process by tuning the hyperparameters in a brute-force fashion.
-	The baselines should be compared, to support the effectiveness of this proposed algorithm. For example, Behl et al. (2019) automatically tuning the learning rates during training definitely needs to be compared, since both aim to stabilize the training of MAML.

**Experience Assessment:**

I have published one or two papers in this area.

**Review Assessment: Checking Correctness Of Derivations And Theory:**

I assessed the sensibility of the derivations and theory.

**Review Assessment: Checking Correctness Of Experiments:**

I carefully checked the experiments.

**Review Assessment: Thoroughness In Paper Reading:**

I read the paper thoroughly.

---

> ### Author Response · Authors · 2019-11-11
> **Response 1**
>
> Thank you for your thoughtful review and comments. Our response to your reviews are as follows:
>
>
> > In Section 3.1.1, the authors ignored Tg based on an observation that Tg is small. However, even in the Appendix, the authors did not explicate the experimental setting where they reach such a conclusion. Will Tg be always small then, in any task and any dataset?
>
> We specified that the experimental setting is the same as that of Section 5.2 except that particular learning rates are used and the total number of iterations is larger in the Appendix of the new manuscript. We confirmed that at least Tg is small under that setting where we conducted the sinusoid regression. Although we did not check the Omniglot and MiniImagenet classification, we believe that the presumption is valid enough for predicting the experimental result since the result of our theory is applicable even for these datasets and tasks.
>
>
> > Practically, we often train MAML in 5 or 10 steps instead of only one step. Will the conclusion apply to such setups? Or will the model itself diverge during the inner loop as more steps are taken, provided with a larger value of \alpha?
>
> Although we do not investigate the multi-step cases, we think that our conclusion has an implication for these cases as well. Ignoring higher-order terms and assuming that our assumptions hold, the regularization term of the simplified meta-objective will be from the squared norm of the gradient to the inner product of gradient of the loss at step 0 and sum of the gradient at each step, as we mentioned in Appendix A.2 : \tilde L(\theta)  = L(\theta) - \alpha / 2 g^T g —> \tilde L(\theta) = L(\theta) - \alpha / 2 \sum_i g_k^T g_i. Intuitively, if all gradients are orthogonal each other, a multi-steps case is not that different from a one-step case. On the contrary, if all of them are in the same direction, the gradient norm will be multiplied by the number of steps, making the largest possible \alpha will be divided by the number of steps. In the real setting, cosine similarity of gradients will be between 0 and 1, the maximum possible \alpha is expected to be a little bit smaller.
>
>
> > Practically, we merely use vanilla SGD. The figure in the Appendix shows that the main conclusion actually does not apply to Adam, which disempower the practicability of the theoretical guideline.
>
> Thank you for your feedback. We acknowledge that it would have been better if our conclusion applied to Adam as well for practicability. Because the effective learning rate of Adam is defined parameter-wise and changes during training, it was not suitable to predict the experimental result based on the conclusion from the theory about a fixed scalar learning rate. One of our contributions is suggesting to regard MAML as optimization with the negative gradient penalty and this formulation itself, of course, can be used for analyzing the adaptive optimizer too, though the procedure of the analysis will not be the same as what we did in this study. Comparing the result with ours may reveal why we seem to get a different result between gradient descent and Adam. Even though our result may not be immediately applicable to Adam, we think that our theory still has the value as the first step to analyze the case including optimizer like Adam.
>
>
> > How can we search the “largest possible” inner loop learning rate? Should we still use grid-search or heuristic search? In that case, what is the implication/shortcut that this proposed guideline brings?
>
> The key takeaway of our work is that you do not have to find the “optimal pair of \alpha and \beta”, which is unknown, to tune the learning rates. Rather, it is enough to “identify the largest possible \alpha” for MAML to converge. Since you just need to find a particular \alpha, you can find the maximum possible \alpha with the binary search algorithm which runs only in logarithmic time even in the worst case, for example. This does not take that much time because you do not have to wait until convergence to find the largest possible \alpha.

---

> ### Author Response · Authors · 2019-11-11
> **Response 2**
>
> > The baselines should be compared, to support the effectiveness of this proposed algorithm. For example, Behl et al. (2019) automatically tuning the learning rates during training definitely needs to be compared, since both aim to stabilize the training of MAML.
>
> Thank you for your feedback. Admitting that both works concern about the stability of MAML training, we think that our work is a different type of study from Behl et al. (2019). Although contribution of Behl et al. (2019) was proposing a specific algorithm to stabilize MAML training, our study focuses on investigating the mechanism of MAML training and revealing when MAML training fails, and our main argument is independent of a specific algorithm. Also, we think that the study of Behl et al. (2019) and our work are hard to be compared. A guideline we suggested “Identifying the large \alpha first and then tuning \beta” is a guideline to search the learning rates for stable training as long as  you use fixed scalar learning rates. Learning rates identified following the guideline are maximum possible fixed learning rates. Therefore, if we compare the training of the learning rates with that of Behl et al. (2019)’s algorithms, it will be just a comparison between the performance of fixed learning rates and that of auto-tuned learning rates and not a comparison between our guideline and Behl et al. (2019)’s algorithms. We appreciate your constructive feedback.

---

### Official Review · AnonReviewer3 · 2019-10-22
**Official Blind Review #3**

**Rating:** 3

**Review:**

The authors study a method to help tuning the two learning rates used in the MAML training algorithm. First, they derive a necessary condition for the convergence of the gradient descent in the single task setting. The condition relies on the eigenvalues of the Hessian of the task loss. This condition is reminiscent of convergence criteria for gradient descent on quadratic objectives, and the authors make the interesting observation that the criteria for the exterior learning rate beta depends on the interior learning rate alpha.
However, in the setting of interest that is the multitask learning, the trick used to analyse the eigenvalues doesn’t work anymore. To circumvent this issue, the authors provides a sufficient condition for the multitask equivalent of the necessary condition to hold.

The derivations are correct (few typos, see nitpick below) and the paper is nicely presented. I’m not a MAML expert so I’ll let the other reviewers judge how this paper compares to the current literature. However, I think that the empirical work can be pushed further. The experiments on Omniglot and MiniImagenet are coherent with the theory, but I am not completely sure of their impact. Indeed, the learning rate domain on which MAML converge seems coherent with what the authors are predicting, but it doesn’t allow us to choose the learning rates before training the model. It doesn’t help that the criteria relies on the spectrum of the Hessian evaluated at a critical point, which of course is not known before convergence in non quadratic setting.
The paper would be way more convincing if the authors could use their findings to describe some more precise heuristics to tune the learning rates, and conduct a proper comparison of its performance with other methods.

As a result, I think the paper is exploring an interesting direction, but that the empirical work might be too preliminary for publication.

Nitpick:

- Notations in 2.2 are a bit confused.The gradient of L wrt theta is denoted first with nabla, then with partial derivatives. Nabla is then used to denote the Jacobian of theta’ wrt theta. There is a small mistake in the 4th line of 2.2. Chain rule for gradient of L wrt theta’ does not give (d L / d theta) times gradient but instead give gradient times (d L / d theta)^T. The error is fixed when passing from (3) to (4) so it doesn’t impact the paper.
- Also, a transpose is missing in the taylor expansion on page 3.
- "A multilayer perceptron with two hidden units of size 40" -> layers?


**Experience Assessment:**

I do not know much about this area.

**Review Assessment: Checking Correctness Of Derivations And Theory:**

I assessed the sensibility of the derivations and theory.

**Review Assessment: Checking Correctness Of Experiments:**

I assessed the sensibility of the experiments.

**Review Assessment: Thoroughness In Paper Reading:**

I read the paper at least twice and used my best judgement in assessing the paper.

---

> ### Author Response · Authors · 2019-11-11
> **Response**
>
> Dear Reviewer 3,
>
> Thank you for your review. we would like to address your concern below.
>
>
> > It doesn’t help that the criteria relies on the spectrum of the Hessian evaluated at a critical point, which of course is not known before convergence in non quadratic setting. The paper would be way more convincing if the authors could use their findings to describe some more precise heuristics to tune the learning rates, and conduct a proper comparison of its performance with other methods.
>
> Thank you for your comment. We do not think that it is essential to know the specific value of the spectrum of the Hessian. Contribution of our work is not only identifying the specific value of the upper bounds but rather revealing the relationship between the upper bounds of the inner learning rate and meta-learning rate. Thanks to the relationship, we can show the guideline suggesting that we should identify the largest possible inner learning rate.

---

### Official Review · AnonReviewer2 · 2019-10-24
**Official Blind Review #2**

**Rating:** 1

**Review:**

This paper studies the two learning rates \alpha and \beta used in Model-Agnostic Meta-Learning (MAML) algorithm by [Finn et al., 2017]. MAML is known to be difficult to train, and part of the reason why is the need to tune the two learning rates. Under simplifications, the paper derives some necessary conditions on \alpha and \beta for the MAML iterates to converge to local minima, and then verifies the theory by experiments on synthetic and real-world data.

Overall, I think this paper should be rejected. In my opinion, this paper analyzes a simplified setting which is far from the original MAML setting and yet derives conditions that are not very meaningful or useful. Although it is interesting to see some match between theory and experiments (especially Fig 4), the theory part seems to have room for improvement, and I will detail the reasons in the following.

I believe that the analysis is done on an overly simplified setting and easily breaks without such simplifications.
- The paper assumes that training set is equal to test set and these sets do not change over iterations, which is essentially equivalent to assuming full gradient access; this is different from the stochastic setting of MAML.
- The "necessary condition" is derived in the case where there is only one task, and there is no discussion on generalizing to some other tasks. This means that the setting is too simplified so that it is not even a meta-learning setup.
- In the extension to multiple tasks, the paper derives "sufficient conditions" to a set of "necessary conditions" for convergence of MAML to local minima. This means that, if we consider the sets of events A = {convergence of MAML to local min}, B = {necessary conditions for A}, and C = {sufficient conditions for B}, A is a subset of B and C is a subset of B and nothing can be specified between A and C. In the extreme case, A and C may be even disjoint subsets of B. Thus, at least in theory, the conditions for multiple tasks do not tell us anything about convergence.
- The analysis relies on a number of "A is approximately equal to B" arguments without careful handling of errors, e.g., (3) and (4), and Tg = 0.

I also have concerns about the correctness of the analysis in the single task case.
- In Section 3.1.2 and eq (12), the paper reparametrizes \theta to v and analyzes updates on v. How is (12) obtained from the original update rule of \theta? In fact, the paper analyzes steepest GD but doesn't provide the explicit update rule; for example, is it steepest GD with respect to which norm? It'd be helpful to have the details in the main text.
- Even if (12) is true, there are pathological cases lying in a set of measure zero, where (12) converges even when (13) is not satisfied. For simplicity, consider v(t+1) = G v(t), where G is a real symmetric matrix. Assume all but one eigenvalue \lambda_1 are greater than 1 and |\lambda_1| < 1. Let v_1 be the corresponding eigenvalue for \lambda_1. Then, if v(0) = v_1, the sequence v(t) converges to zero. This means that the "necessary condition" derived in (16) may not actually be a necessary condition for convergence.

There are also some points in the main text that doesn't describe the setting clearly; if the reader has no prior knowledge of MAML algorithm, they may get confused.
- In the update rule (1) and (2), it'd be better to mention that \nabla_\theta L_\tau (\theta) is an *estimate* of the true gradient using the training data and test data, respectively. In their current status, the gradients in (1) and (2) will read as the full gradients of the population where the training/test data points are sampled from.
- Section 2.1 only introduces the case where the update (1) is done only once in the inner loop. However, later Section 2.2 and Section 3, the paper says "only one step is taken for update...". Without the prior knowledge that there are versions in which multiple updates (1) are done, the readers can easily get confused.

Minor comments
- The word "minima" used throughout should better be corrected to "local minima," as mere minima may be understood as "global minima."
- In the abstract, there is a phrase "in contrast to the case of using the normal gradient descent method." Which setting do you mean by the "normal gradient descent"?
- After equations (3) and (4), the paper says "The above is known as the first-order approximation...", but as far as I'm concerned, the first-order approximation version of MAML is (3) without the hessian term, not (4). This statement can potentially be misleading.
- Column vectors / row vectors are mixed up. In (3), the partial derivatives are row vectors, but in (4) g_\tau(\theta)'s are column vectors. However, right above (3), g_\tau(\theta) is a row vector this time.
- Did (5) come from the approximation (4)? If so, as \tilde L(\theta) is defined to be L(\theta'), the two sides of (5) are "approximately equal" not "equal."
- In Eq (19): P_\tau (\theta - \theta^*) -> (\theta-\theta^*).
- At the end of page 6, what do you mean by "not a sum but a mean"? Isn't sum of n things equal to the mean, except for a factor of 1/n?

**Experience Assessment:**

I have read many papers in this area.

**Review Assessment: Checking Correctness Of Derivations And Theory:**

I assessed the sensibility of the derivations and theory.

**Review Assessment: Checking Correctness Of Experiments:**

I assessed the sensibility of the experiments.

**Review Assessment: Thoroughness In Paper Reading:**

I read the paper at least twice and used my best judgement in assessing the paper.

---

> ### Author Response · Authors · 2019-11-11
> **Response 1**
>
> Dear Reviewer 2,
>
> Thank you for your careful review and constructive feedback. We admit that it was inaccurate and misleading to say that “we derived the necessary condition for MAML to converge to minima”. Considering your feedback, we changed the expression to “we derived a condition for a simplified MAML to locally converge to local minima from any point in the vicinity of the local minimum” since it is more accurate to explain what we did. We also modified our manuscript in consideration of your other comments. We appreciate your insightful feedback. Though we modified our theory, we think that the implication from our theory itself is still useful and is not reduced by the modifications. We would like to respond to your comment one by one.
>
>
> > The paper assumes that training set is equal to test set and these sets do not change over iterations, which is essentially equivalent to assuming full gradient access; this is different from the stochastic setting of MAML.
>
> As you point out, we do not consider the stochasticity of the data sampling from the same task, though task during training can be different from that for fine-tuning, or course. We discuss the averaged behavior of a simplified MAML throughout the training in the manuscript. Admitting that stochasticity of the data from the same task matters and it would be better to consider it to derive the condition for a more realistic case, looking at the average case still often tells us important implications as our theory predict the result of the experiment regardless of such a simplification. We believe that our research is a good first step to analyze the relationship between the inner learning rate \alpha and the meta-learning rate \beta further. As to the variability between tasks, we will respond later.
>
>
> > The "necessary condition" is derived in the case where there is only one task, and there is no discussion on generalizing to some other tasks. This means that the setting is too simplified so that it is not even a meta-learning setup.
>
> As we mentioned, it was inaccurate to say that we derived the necessary condition for MAML to converge to minima and we changed the expression. We admit that the one-task setting is too simple to call MAML. As you comment, it is not any more multi-task learning. Yet, this simplification allows us to find the relationship between the inner learning rate and meta-learning rate and in the sense, this simplification is useful for researchers to acquire an intuitive understanding of how MAML works.
>
>
> > In the extension to multiple tasks, the paper derives "sufficient conditions" to a set of "necessary conditions" for convergence of MAML to local minima. This means that, if we consider the sets of events A = {convergence of MAML to local min}, B = {necessary conditions for A}, and C = {sufficient conditions for B}, A is a subset of B and C is a subset of B and nothing can be specified between A and C. In the extreme case, A and C may be even disjoint subsets of B. Thus, at least in theory, the conditions for multiple tasks do not tell us anything about convergence.
>
> We admit that in the extreme case, A and C may be disjoint. Considering your point out, we altered A = {convergence of MAML to local min} to A’ = {local convergence of a simplified MAML to local min from any point in the vicinity of local minimum}. Therefore, the “necessary condition” we originally derived is the necessary and sufficient condition and the condition of multi-task setting is now the sufficient condition of A’. Also, since we can talk about multi-task setting thanks to this modification, we are able to consider the variability between different tasks. We think that this grasps the essence of MAML as the conclusion of our theory is consistent with the experimental result. Thank you for your feedback．
>
>
> > The analysis relies on a number of "A is approximately equal to B" arguments without careful handling of errors, e.g., (3) and (4), and Tg = 0.
>
> As we mentioned before, the “condition of MAML” was misleading. It is more accurate to call our condition “condition of a simplified MAML” in consideration of the approximation you pointed out as well. We think that errors from the approximation are negligible at least to investigate the relationship between \alpha and \beta since our theory can explain well the result of experiments with the benchmark dataset.
>
>
> > In Section 3.1.2 and eq (12), the paper reparametrizes \theta to v and analyzes updates on v. How is (12) obtained from the original update rule of \theta? In fact, the paper analyzes steepest GD but doesn't provide the explicit update rule; for example, is it steepest GD with respect to which norm? It'd be helpful to have the details in the main text.
>
> Thank you for your feedback. we added the details in the main text and corrected a mistake too: P(\theta - \theta*) —> P^T(\theta - \theta*).

---

> > ### Comment · AnonReviewer2 · 2019-11-14
> > **Response Acknowledged**
> >
> > Dear Authors,
> >
> > Thanks for the detailed response. I have also checked the updated manuscript.
> >
> > I still have some questions regarding the updates:
> >
> > - In our updated statement "we derived a condition for a simplified MAML to locally converge to local minima from any point in the vicinity of the local minimum," is the condition necessary or sufficient? Judging from the response, I think the authors meant sufficient condition. However, the main text still uses expressions such as "the condition that $\alpha$ should satisfy for $\theta^*$ to be a local minimum..." and "the condition that $\beta$ must satisfy is as follows..." which mean necessary conditions.
> >
> > Also, note that, (eigenvalues of Hessian) ≥ 0 is a *necessary* condition for local minima, whereas (eigenvalues of Hessian) > 0 is a *sufficient* condition. This means that in Section 3.1.1, eq (9), the authors are still writing a *necessary* condition for local minima, while in Section 3.1.2, the authors seem to derive a *sufficient* condition for convergence. I'm still confused.
> >
> > - To clarify: by your response to the multi-task case, do you mean that your condition is now a sufficient condition for local convergence to a local minimum at its vicinity?

---

> > > ### Author Response · Authors · 2019-11-15
> > > **Thank your for your response**
> > >
> > > Dear Reviewer 2,
> > >
> > > Thank you for your polite and quick response. We thought we had derived the necessary and sufficient condition for the single-task case and a sufficient condition for the multi-task case. However, as you appropriately point out, (eigenvalues of Hessian) ≥ 0 is a necessary condition and  (eigenvalues of Hessian) > 0 is a sufficient condition. Hence, what we said in the manuscript did not explain what we actually did correctly. That was our oversight. We also acknowledge that our expression was still confusing. Therefore, we re-updated the manuscript to make it clear that we derived a sufficient condition for both the single-task case and multi-task case. Accordingly, we corrected all expressions to be consistent with this modification, i.e. condition → sufficient condition. We uploaded the revised manuscript. We appreciate again your sincere contribution to the constructive discussion.

---

> ### Author Response · Authors · 2019-11-11
> **Response 2**
>
> > Even if (12) is true, there are pathological cases lying in a set of measure zero, where (12) converges even when (13) is not satisfied. For simplicity, consider v(t+1) = G v(t), where G is a real symmetric matrix. Assume all but one eigenvalue \lambda_1 are greater than 1 and |\lambda_1| < 1. Let v_1 be the corresponding eigenvalue for \lambda_1. Then, if v(0) = v_1, the sequence v(t) converges to zero. This means that the "necessary condition" derived in (16) may not actually be a necessary condition for convergence.
>
> We admit that we were not able to derive the “necessary condition” in the original paper. We changed the expression from just “convergence” to “local convergence from any point in the vicinity of local minimum” to eliminate the pathological case that you showed.
>
>
> > In the update rule (1) and (2), it'd be better to mention that \nabla_\theta L_\tau (\theta) is an *estimate* of the true gradient using the training data and test data, respectively. In their current status, the gradients in (1) and (2) will read as the full gradients of the population where the training/test data points are sampled from.
>
> We articulated in the new manuscript that \nabla_\theta L_\tau (\theta) is an estimate of the true gradient as you appropriately advised.
>
>
> > Section 2.1 only introduces the case where the update (1) is done only once in the inner loop. However, later Section 2.2 and Section 3, the paper says "only one step is taken for update...". Without the prior knowledge that there are versions in which multiple updates (1) are done, the readers can easily get confused.
>
> We specified in the new manuscript that MAML allows multiple steps in Section 2.1.
>
>
> > The word "minima" used throughout should better be corrected to "local minima," as mere minima may be understood as "global minima."
>
> We changed all ”minima” to ”local minima”．
>
>
> > In the abstract, there is a phrase "in contrast to the case of using the normal gradient descent method." Which setting do you mean by the "normal gradient descent"?
>
> We meant the normal gradient descent as the training when \alpha =0. We admit it was confusing expression and deleted the phrase.
>
>
> > After equations (3) and (4), the paper says "The above is known as the first-order approximation...", but as far as I'm concerned, the first-order approximation version of MAML is (3) without the hessian term, not (4). This statement can potentially be misleading.
>
> "The above is known as the first-order approximation..." was misleading and could give expression as if we did the well-known first-order approximation. We articulated that the result with our approximation can be regarded as the result of well-known first-order approximation only when all of our assumptions hold.
>
>
> > Column vectors / row vectors are mixed up. In (3), the partial derivatives are row vectors, but in (4) g_\tau(\theta)'s are column vectors. However, right above (3), g_\tau(\theta) is a row vector this time.
> > Did (5) come from the approximation (4)? If so, as \tilde L(\theta) is defined to be L(\theta'), the two sides of (5) are "approximately equal" not "equal."
> > In Eq (19): P_\tau (\theta - \theta^*) -> (\theta-\theta^*).
>
> Following your advice, we corrected all of them.
>
>
> > At the end of page 6, what do you mean by "not a sum but a mean"? Isn't sum of n things equal to the mean, except for a factor of 1/n?
>
> Whether the loss is sum or average affects the value of the apparent learning rate. If we use the summation loss, the apparent learning rate seems to be n-times larger than when the averaged loss is used. In our research, we would like to discuss how large the maximum learning rate can be. Though we write the formulation of MAML in Section 2 following the formulation in the original MAML paper, we would like to make learning rates independent of the number of tasks and the sample size and hence we experimented with the average loss.

---

### Author Response · Authors · 2019-11-11
**Summary of the overall revision**

Dear reviewers,

Thank you for your detailed review and constructive feedback. We uploaded the revised manuscript. Although we responded to each reviewer, we would like to explain a summary of the revision below.

- Considering the reviewer 2’s feedback, we changed the expression “the necessary condition for MAML to converge to minima” to “the condition for a simplified MAML to locally converge to local minima from any point in the vicinity of the local minima” because this is more suitable expression to explain what we did. Hence, we altered all expressions to expressions consistent with this modification, i.e. convergence —> local convergence.

- We added the detail derivation of Eq. (12) in the main text.

- We explicitly write in Appendix C that the calculation of Tg is done for the training loss of the sinusoid regression and the experimental setting is the same as 5.2 except that the total number of iterations is 50000 and learning rates \alpha and \beta are fixed to be 1e-2 and 1e-3 respectively.

- We altered all minor mistakes and misleading expressions pointed out, i.e. we indicate that \nabla_\theta L_\tau (\theta) is an *estimate* of the true gradient.

---

### Decision · Program_Chairs · 2019-12-19

**Decision:**

Reject

**Comment:**

This paper theoretically and empirically studies the inner and outer learning rate of the MAML algorithm and their role in convergence. While the paper presents some interesting ideas and add to our theoretical understanding of meta-learning algorithms, the reviewers raised concerns about the relevance of the theory. Further the empirical study is somewhat preliminary and doesn't compare to prior works that also try to stabilize the MAML algorithm, further bringing into question its usefulness. As such, the current form of the paper doesn't meet the bar for ICLR.